# Prebiotic chiral transfer from self-aminoacylating ribozymes may favor either handedness

Josh Kenchel [1,2,3,9], Alberto Vázquez-Salazar [3,9], Reno Wells [2], Krishna Brunton [2], Evan Janzen[1,2], Kyle M. Schultz[4], Ziwei Liu [5,6], Weiwei Li[7], Eric T. Parker[8], Jason P. Dworkin [8] & Irene A. Chen [1,2,3,4] ✉

Modern life is essentially homochiral, containing D-sugars in nucleic acid backbones and L-amino acids in proteins. Since coded proteins are theorized to have developed from a prebiotic RNA World, the homochirality of L-amino acids observed in all known life presumably resulted from chiral transfer from a homochiral D-RNA World. This transfer would have been mediated by aminoacyl-RNAs defining the genetic code. Previous work on aminoacyl transfer using tRNA mimics has suggested that aminoacylation using D-RNA may be inherently biased toward reactivity with L-amino acids, implying a deterministic path from a D-RNA World to L-proteins. Using a model system of self-aminoacylating D-ribozymes and epimerizable activated amino acid analogs, we test the chiral selectivity of 15 ribozymes derived from an exhaustive search of sequence space. All of the ribozymes exhibit detectable selectivity, and a substantial fraction react preferentially to produce the D-enantiomer of the product. Furthermore, chiral preference is conserved within sequence families. These results are consistent with the transfer of chiral information from RNA to proteins but do not support an intrinsic bias of D-RNA for L-amino acids. Different aminoacylation structures result in different directions of chiral selectivity, such that L-proteins need not emerge from a D-RNA World.

All known life is homochiral, consisting almost exclusively of right-handed (D-) sugars and left-handed (L-) amino acids[1]. Indeed, homo-chirality is considered to be a hallmark of life itself, and enantiomeric ratios of organic monomers are used as a biosignature informing the search for life elsewhere[2,3]. The selective advantages of homochirality, namely structural specificities that confer chemical efficiencies, could drive natural selection toward a system with nearly complete chiral asymmetry[4]. Therefore, factors determining the direction of the imbalance, and the exact mechanisms responsible for creating, ampli-fying, and perpetuating the imbalance, are important subjects of study[5].

The emergence of homochirality from a racemic abiotic milieu can be considered in three stages: symmetry breaking, in which the imbalance is introduced; amplification, in which the initial imbalance increases until homochirality is reached; and chiral transfer, in which one homochiral molecular type (e.g., RNA) passes the imbalance to another type (e.g., peptides), breaking the symmetry of the latter[6].

[1]Program in Biomolecular Science and Engineering, University of California, Santa Barbara, CA, USA. [2]Department of Chemistry and Biochemistry, University of California, Santa Barbara, CA, USA. [3]Department of Chemical and Biomolecular Engineering, University of California, Los Angeles, CA, USA. [4]Department of Chemistry and Biochemistry, University of California, Los Angeles, CA, USA. [5]Department of Earth Sciences, University of Cambridge, Cambridge, USA. [6]MRC – Laboratory of Molecular Biology, Francis Crick Avenue, Cambridge Biomedical Campus, Cambridge CB2 0QH, United Kingdom. [7]Bren School of Environmental Management, University of California, Santa Barbara, CA, USA. [8]Solar System Exploration Division, NASA Goddard Space Flight Center, Greenbelt, MD, USA. [9]These authors contributed equally: Josh Kenchel, Alberto Vázquez-Salazar. ✉e-mail: ireneachen@ucla.edu

While recent research has substantially advanced the understanding of symmetry breaking and chiral amplification[7–9], chiral transfer is still not understood well.

The RNA World hypothesis posits that at some point on the early Earth, life went through a stage in which RNA played both roles of information carrier and catalyst in primitive living systems[10–12]. Since modern life consists of D-sugars, the RNA World is assumed to have shared this handedness. Interestingly, enantioenrichment of ribonucleotide precursors can be promoted by amino acids and dipeptides during abiotic synthesis, indicating the potential for chiral transfer between small molecules even before the RNA world became established[13,14]. The emergence of protein translation in the RNA World represents a major transition, during which the chiral nature of RNA could have been transferred to coded proteins. A critical bridge during this transition would have been the 'adapter' molecules that matched individual codons with specific amino acids. While tRNAs, charged by aminoacyl-tRNA synthetases, perform the role of adapter molecules today, self-aminoacylating ribozymes may have been the RNA World precursors of this system[15–18]. Given the modern observation that nucleic acids have D- sugars and proteins have L- amino acids, it has been hypothesized that D-RNA might have inherent affinity or reaction with L-amino acids or their precursors, and therefore a D-RNA World would lead deterministically to a L-protein world (reviewed in ref. 19). Indeed, earlier studies suggested that D-RNA would favor L-amino acids, based on the observation that an RNA minihelix mimicking the acceptor stem of tRNA reacts more quickly with an aminoacyl phosphate oligonucleotide when the aminoacyl chirality is L- rather than D-[20,21]. Furthermore, aminoacyl transfer between strands of a model tRNA acceptor stem also showed stereoselectivity preferring the L-enantiomer in two different transfer mechanisms[22,23]. These studies on structural tRNA mimics suggest an intrinsic reactivity of D-RNA for L-amino acids. Additional studies have also suggested a bias for non-covalent binding interactions between D-RNA and L-amino acids[24,25]. These lines of evidence are consistent with the appealing hypothesis that a D-RNA World is predisposed to give rise to L-proteins.

On the other hand, some variants of an aminoacyl transferase ribozyme (the flexizyme) are capable of promiscuous activity, including catalyzing reactions between tRNA mimics and D-substrates[26], indicating the potential for chiral transfer from D-ribozymes to D-peptides. However, because the flexizyme was originally selected for reaction with L-substrates[27], these observations cannot directly address whether D-ribozymes would have an intrinsic preference for L-versus D- substrates. Overall, some evidence supports a bias of D-RNA toward L-amino acids, but the hypothesis that a D-RNA World would be intrinsically predisposed to lead to an L-peptide world requires further testing. In particular, in an alternative RNA World utilizing different catalytic structures for aminoacylation, would D-RNA still favor reaction of L-amino acids?

In previous work, several self-aminoacylating ribozymes were discovered through in vitro selection for reaction with an activated amino acid, biotinyl-Tyr(Me)−5(4H)-oxazolone[17,28]. The 5(4H)-oxazolones can be synthesized under prebiotically relevant conditions and are known to react to form peptides as well as aminoacylated nucleotides[28–32]. The oxazolone substrates epimerize at the α-carbon due to tautomerization[31,33], so both L- and D- chirality at the α-carbon are represented in the reaction in equal amounts. The starting library of this selection had been a pool of RNA containing a random sequence flanked by regions of constant sequence, where the random region had nearly complete coverage of sequence space[17]. The ribozymes catalyze the formation of an ester bond with an internal 2′-OH to form aminoacyl-RNA, fixing the chirality of the α-carbon upon reaction. Three distinct sequence motifs, having distinct aminoacylation sites, emerged from the selection, including several subfamilies and many individual ribozymes. All ribozyme motifs were found to react at internal 2′-OH sites (i.e., not at a strand terminus), unlike the modern tRNA acceptor stem and previously studied systems. The ribozymes also tolerate some changes to the substrate side chain, particularly for aromatic side chains[34]. The unusual reactive sites on the RNA, as well as the fact that the ribozymes were discovered through an exhaustive search of sequence space, indicate that these ribozymes are orthogonal to natural and tRNA-inspired systems. This orthogonality offers an opportunity to test whether the propensity for D-RNA to react to give L-aminoacyl products is in fact generalizable.

In this work, we analyze the ratio of D- to L- aminoacyl products in the reactions between a representative set of these ribozymes and the substrate biotinyl-Phe-5(4H)-oxazolone (BFO)[34], to test the existence of an intrinsically biased stereoselectivity governing chiral transfer from D-RNA to L-amino acids.

## Results

### Ribozyme reactions with biotinyl-Phe-oxazolone substrate

Prior work from an exhaustive search of sequence space (random region of 21 nucleotides) using sequencing to measure catalytic activity paired with in vitro evolution (SCAPE) had identified three ribozyme sequence motifs[17]. These ribozymes catalyze self-aminoacylation with biotinylated 5(4H)-oxazolones with aromatic side chains[34]. Ten ribozymes representing a diverse set of the most active sequences from the three major motifs had been previously validated for activity (Table 1). These included the most abundant representative from each major family or subfamily (eight sequences designated by the "-a" suffix), as well as two other sequences with significant activity (one from Motif 1 and one from Motif 2). In addition, we chose five more sequences from the most active family (Motif 2). These were selected by first using EMSA to assay the activity of 12 candidate sequences from Motif 2, for which previous high-throughput sequencing had suggested high activity[34]. Of those 12 candidates, we chose the five ribozymes with the greatest self-aminoacylation by biotinyl-Phe-oxazolone (BFO), determined by EMSA, to include in the present analysis (Supplementary Fig. 1; Table 1). Thus, a total of 15 sequences (six from Motif 1, eight from Motif 2, and one from Motif 3) were tested in the following experiments.

These ribozymes were originally identified by reaction with biotinyl-Tyr(Me)-oxazolone (BYO) and were subsequently shown to react with substrates having aromatic side chains, including BFO[17,34]. BFO is an ideal substrate for chiral study since methods for separating and detecting enantiomers of phenylalanine have been previously established[35–37]. BFO was chemically synthesized and the product was verified by proton NMR (Supplementary Figs. 2–3). As a 5(4H)-oxazolone, BFO is subject to epimerization, and the rate of degradation by hydrolysis was expected to be similar for both stereoisomers. Therefore, the BFO substrate was expected to contain a 1:1 mixture of both stereoisomers. To verify this, samples of BFO (500 nM) were hydrolyzed by mildly basic and then strongly acidic conditions to yield Phe (L and D). The ratio of L-Phe to D-Phe was not statistically significantly different from 1:1 using chiral chromatography (Supplementary Fig. 4).

### Ribozymes display a range of stereoselectivities

We measured the stereoselectivity of the 15 ribozymes by conducting overnight aminoacylation reactions with different concentrations of substrate BFO. The amounts of L and D isomers incorporated during aminoacylation were determined after each reaction, by mild basic hydrolysis of aminoacylated RNA to release biotinyl-Phe, followed by acid hydrolysis to release Phe (L and D) (Supplementary Fig. 5). Hydrolysis of the aminoacyl ester bond was essentially quantitative, with no aminoacyl-RNA detectable by EMSA after hydrolysis (Fig. 1A, Supplementary Fig. 6). LC-MS with a chiral column was used to quantify L-Phe and D-Phe (Fig. 1B–F). A more sensitive detection protocol involving derivatization was also used to validate the results, and results from both protocols correlated well (Supplementary Fig. 4).

**Table 1 | Ribozyme sequences analyzed in this work**

| Name | Reference | RNA sequence of variable region | Motif | $i_e$ | % $ee_D$ | $k$ (M$^{-1}$min$^{-1}$) Refs. 34,57 | $A$ Refs. 34,57 |
|---|---|---|---|---|---|---|---|
| S-1A.1-a | 17 | CUACUUCAAACAAUCGGUCUG | 1 | 2.13 (1.53, 2.74) | 62.56 % [19.76] | 118.06 (20.17, 181.83) | 0.79 (0.45, 1.00) |
| S-1A.1-n | 17 | CUCUUCAAACAAUCGGUCUUC | 1 | 1.32 (1.18, 1.49) | 41.73 % [4.85] | 122.46 (47.98, 227.55) | 0.27 (0.22, 0.46) |
| S-1B.1-a | 17 | CCACACUUCAAGCAAUCGGUC | 1 | 2.03 (1.40, 3.35) | 63.28 % [22.12] | 87.49 (34.90, 196.82) | 0.99 (0.62, 1.00) |
| S-1B.2-a | 17 | CCGCUUCAAGCAAUCGGUCGC | 1 | 3.54 (2.88, 4.02) | 86.35 % [8.76] | 136.11 (48.96, 279.25) | 0.29 (0.21, 0.50) |
| S-1B.3-a | 17 | CCGAGUUUCAAGCAAUCGGUC | 1 | 2.93 (2.91, 2.95) | 91.91 % [0.32] | 320.34 (112.17, 592.19) | 0.61 (0.48, 0.86) |
| S-1C.1-a | 17 | CUCUUCAAUAAUCGGUUGCGU | 1 | 1.63 (1.06, 1.97) | 53.65 % [15.82] | 103.94 (13.09, 503.88) | 0.41 (0.14, 1.00) |
| S-2.1-a | 17 | AUUACCCUGGUCAUCGAGUGA | 2 | −0.69 (−1.38, −0.07) | −22.53 % [32.33] | 1,656.27 (146.41, 4,616.20) | 0.55 (0.37, 0.80) |
| S-2.1-t | 17 | AUUACCCUGGUCAUCGAGUGU | 2 | −1.98 (−2.35, −1.42) | −64.02 % [14.83] | 1,860.97 (431.66, 5,167.17) | 0.58 (0.40, 0.81) |
| S-2.2-a | 17 | AUUCACCUAGGUCAUCGGGUG | 2 | −1.3 (−1.66, −1.00) | −43.52 % [11.12] | 2,114.88 (880.72, 5,739.42) | 0.31 (0.21, 0.42 |
| S-3.1-a | 17 | AAGUUUGCUAAUAGUCGCAAG | 3 | −1.03 (−1.14, −0.85) | −34.59 % [5.55] | 65.88 (17.87, 309.25) | 0.66 (0.23, 1.00) |
| P4-1 | 34 | AUUACCUUGGUCAUCGAGUGA | 2 | −0.33 (−0.84, −0.26) | −13.14 % [17.29] | 108.30 (20.69, 2,219.98) | 0.096 (0.042, 0.20) |
| P4-2 | 34 | AUUACCUUGGUCAUCGAGUGU | 2 | −0.36 (−0.48, −0.16) | −13.65 % [6.97] | 49.597 (2.788, 1,428.20) | 0.10 (0.03, 0.99) |
| P4-3 | 34 | AUUACCUAGGUCAUCGAGUGU | 2 | −0.76 (−0.89, −0.51) | −27.39 % [8.29] | 550.48 (212.40, 2,646.26) | 0.44 (0.19, 1.00) |
| P2-4 | 34 | AUUCACCUAGGUCAUCGAGUGU | 2 | −2.20 (−2.40, −1.98) | −67.73% [5.40] | 378.20 (172.18, 612.16) | 0.47 (0.40, 0.61) |
| P1-5 | 34 | AUUCACCUAGGUCAUCGAGUUU | 2 | −2.61 (−2.68, −2.55) | −72.73% [1.11] | 222.78 (33.21, 673.91) | 0.51 (0.26, 1.00) |

Sequence motifs were previously described[17]. Enantioselectivity index $i_e$ = log$_2$($k_D/k_L$) determined by best fit to the first-order rate law, and 95% confidence intervals were determined by bootstrapping and given in parentheses (see "Methods"). Catalytic rate $k$ and maximum aminoacylation fraction $A$ were determined by kinetic sequencing ($k$-Seq)[34,57]. Percent enantiomeric excess (% $ee_D$) was calculated as [(% D-Phe − % L-Phe)/(% D-Phe + % L-Phe)] * 100, with standard deviation given in brackets, for 1 mM substrate concentration. For example, analysis of the single HPLC trace shown in Fig. 1D for S-1A.1-a gave values of D-Phe = 73.6% and L-Phe = 26.4%, giving % $ee_D$ = 47.2%. Note that the % $ee_D$ given in Table 1 is the average of multiple replicates. $ee_D$ did not depend significantly on substrate concentration (Supplementary Fig. 7). Source data are provided as a Source Data file.

The amount of RNA aminoacylated by each stereoisomer over a substrate concentration series was used to calculate the ratio of catalytic rates ($k_D/k_L$) (Fig. 2), according to a pseudo-first-order rate law with excess substrate, given a 1:1 ratio of BFO isomers (Supplementary Note 1). The enantioselectivity index ($i_e$) of a ribozyme sequence was defined as log$_2$($k_D/k_L$), such that a negative value of $i_e$ indicates bias favoring the L isomer, and a positive value of $i_e$ indicates a bias favoring the D isomer. Of the ribozymes tested, nearly half (6/15) showed preference for the D isomer of BFO, with four of those displaying ≥ 2-fold bias). Of the nine L-selective ribozymes, five displayed ≥ 2-fold bias (Fig. 3).

**Stereoselectivity is consistent within sequence families**
The 15 ribozymes represented three sequence motifs[17], with two Motifs (1 and 2) represented by multiple distinct ribozymes. Within each motif, the direction of selectivity was consistent, with all members of Motif 1 preferring D-Phe and all members of Motif 2 preferring L-Phe (Fig. 4A). The difference in enantioselectivity between two ribozymes was generally greater when comparing ribozymes from different motifs (up to $\Delta i_e \approx 6$) rather than ribozymes from the same motif (up to $\Delta i_e \approx 3$) (Fig. 4B).

**Stereoisomer preference is attributable to the aminoacyl α-carbon**
Because BFO contains a biotin group, which itself has chiral centers (Fig. 1B), it was possible that the measured ribozyme preferences might be dependent on stereocenters other than the aminoacyl α-carbon. In theory, this possibility could be tested by reversing the stereocenters in the biotin moiety of BFO and assaying ribozyme

selectivity, or, equivalently, by assaying the selectivity in a reaction of BFO with an L-ribozyme. Ribozyme S-1A.1-a (D-RNA) preferred D-Phe, yielding 80% aminoacylation (95% confidence interval (CI): 77–83%; normality assumed) at [BFO] = 1 mM. The enantioselectivity index ($i_e$) could be quantified in two ways (Supplementary Note 1, Supplementary Fig. 7), with similar results. Fitting data to the rate law over multiple [BFO] reactions gave an estimated $i_{e,D-RNA}$ of 2.07 (95% CI: 1.32–2.77 from bootstrapping (see Methods)). Averaging triplicate measurements at [BFO] = 1 mM gave an estimated $i_{e,D-RNA}$ of 2.7 (95% CI: 1.9–3.5; normality assumed). The second method was preferred due to its simplicity. If the ribozyme's activity is independent of the stereocenters of biotin, then the L- enantiomer of S-1A.1-a is expected to prefer L-Phe, with $i_{e,L-RNA} = -i_{e,D-RNA}$. Indeed, the reaction of the chemically synthesized L-RNA showed a preference for L-Phe, with $i_e = -3.3$ (95% CI: −3.9 to −2.7; normality assumed) (Fig. 4C). The total amount of aminoacylation for L-RNA was 79 ± 3% at 1 mM [BFO] (95% CI: 76–82%; normality assumed), similar to the amount observed for D-RNA, as expected. These results are consistent with the primary determinant of selectivity being the chirality at the α-carbon, rather than biotin stereocenters.

## Discussion
While a significant body of work supports the idea that certain aminoacyl transfer reactions and binding interactions show chiral selectivity in agreement with that found in extant biology (namely, D-RNA favoring interaction with L-amino acids), whether this selectivity would be generalizable to an alternative RNA world has been unclear. The

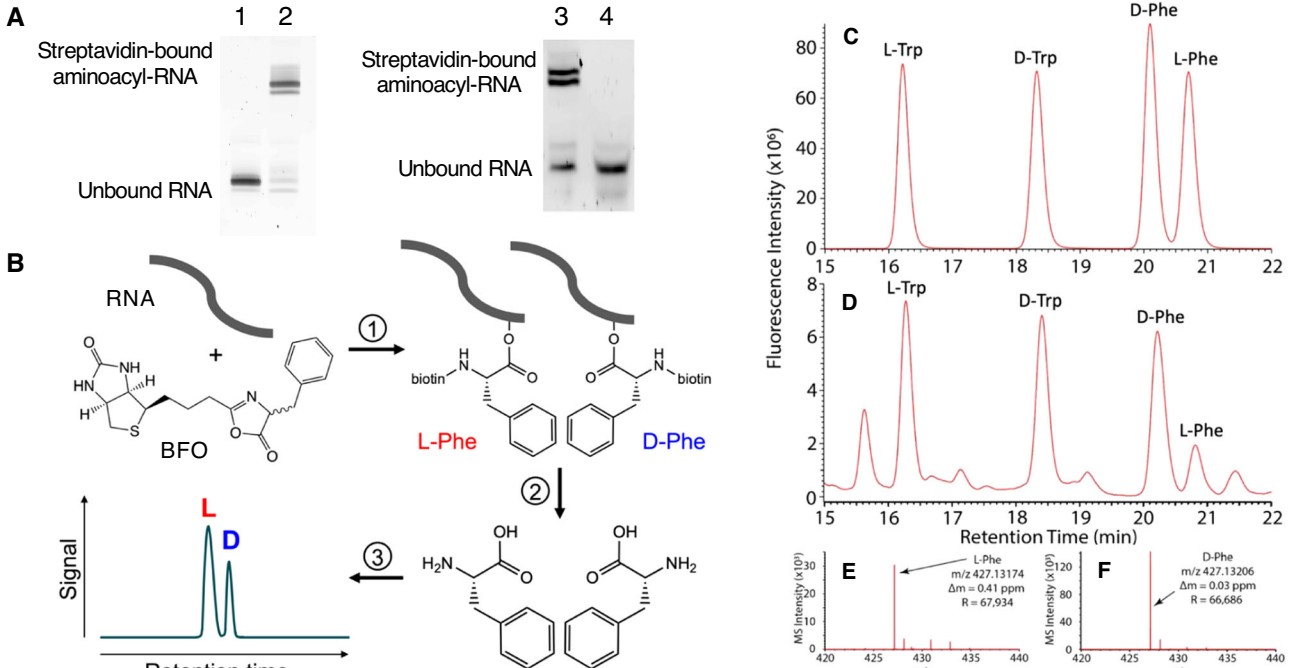

**Fig. 1 | Ribozyme reactivity and analysis of products. A** Example EMSAs of ribozyme S-1A.1-a incubated with streptavidin for gel shift (representative of independent triplicate results). Left gel: before (lane 1) and after (lane 2) reaction with BFO. Right gel: before (lane 3) and after (lane 4) selective hydrolysis of the aminoacyl ester bond. A low mobility band is observed for aminoacyl-RNA after reaction with BFO and is not observed after selective hydrolysis, as expected. **B** To determine stereoselectivity of ribozymes, each ribozyme was incubated with racemic BFO. Upon reaction (1), an ester bond is formed at an internal 2′ hydroxyl to yield aminoacyl-RNA of unknown isomeric ratio. (2) The ester is selectively hydrolyzed, yielding biotinylated phenylalanine, which is separated from the RNA by size exclusion chromatography. The amide bond is then cleaved to remove the biotin moiety. (3) The resulting mixture of L- and D-phenylalanine is separated by HPLC on a chiral column, the peak identities verified by mass spectrometry, and the enantiomeric ratio determined by peak integration. **C, D** Example LC-MS analysis shown for a combined mix of 10 μM standards (**C**) and for the reaction products of S-1A.1-a and BFO (1 mM), after workup (**D**). Racemic Trp was used as an internal standard during ion exchange chromatography. An excess of D-Phe over L-Phe was observed in the sample. Mass spectrometry verified the identity of L-Phe (**E**) and D-Phe (**F**) peaks from the sample used for (**D**). BFO = biotinyl-Phe-5(4H)-oxazolone; RNA = ribonucleic acid.

ribozymes studied here catalyze aminoacylation at internal sites using a prebiotically plausible, epimerizable substrate (5(4H)-oxazolones) and thus represent an orthogonal system compared to tRNA-inspired structures. Although other amino acids may have been more abundant prebiotically[38], phenylalanine has been found in meteorites[39], and multiple different ribozyme motifs have appreciable activity with the BFO substrate[34]. This allows a comparison of chiral preference of unrelated ribozymes, but this reaction is nevertheless an experimental model for a prebiotic reaction. Several of the D-ribozymes analyzed in this work, specifically those from ribozyme Motif 1, were found to be selective for the D-Phe over L-Phe substrate analog. This finding contradicts the hypothesis that D-RNA has an intrinsic general preference for L-amino acids. Instead, chiral preference appears to depend on the specifics of the RNA system.

The homochiral L-protein world observed on Earth today may have arisen from a homochiral D-RNA World, given the important advantages of homochirality, such as greater efficiency of RNA replication[40] and improved peptide folding[41–44]. The results from our model system suggest that either outcome (D-protein or L-protein) would have been possible, depending on the structures used for aminoacylation in the D-RNA world. The observed selectivities are not large (generally < 12-fold), but small imbalances could have been subsequently amplified by natural selection or chemical or physical mechanisms, such as preferential formation of homochiral peptides[5,33]. In addition, enantiomeric excess in the amino acid substrates, stemming from symmetry breaking or amplification processes, could also contribute to the chiral outcome[45]. Chiral preference was consistent within the ribozyme motifs studied, so evolution of aminoacylation ribozymes would not affect the direction of imbalance so long as the general motif was conserved.

A different chemical origin of life may be governed by different principles. Nevertheless, in the context of chiral transfer in an RNA World, the results suggest that when replaying the Gouldian "tape of life"[46] from a homochiral D-RNA world, a homochiral L-protein world would not be guaranteed by a general chemical bias of RNA. Given the likely importance of chance in ribozyme discovery[17,47], the ultimate emergence of an L- vs. D-protein world may have been as random as a coin flip.

## Methods

### Ribozyme sequences and substrate

For each of the fifteen D-ribozyme sequences investigated in this paper, chemical synthesis (Integrated DNA Technologies, PAGE-purified) was used to obtain single-stranded template DNA with the following sequence: 5′-<u>GATAATACGACTCACTATAGGGAATGGATCCACATCTACG AATTC</u>-N_x-TTCACTGCAGACTTGACGAAGCTG-3′, where $N_x$ denotes the variable region and nucleotides upstream of the transcription start site are underlined. Individual sequences are given in Table 1. The ssDNA template was converted to dsDNA by Phusion polymerase (New England Biolabs) and a reverse primer of sequence 5′-CAGCTTCGTCAAGTCTG CAGTGAA-3′, in a 50 μL mixture containing 1.5 μg template, 10 μM primer, 1 mM dNTPs, and 2 U enzyme in 1X HF buffer supplied by the manufacturer, with 68 °C annealing temperature and 72 °C synthesis temperature for 30 min.

RNA was transcribed from the dsDNA reaction mixture using T7 RNA Polymerase (New England Biolabs), in a 30 μL mixture containing 1 μg template, 6.7 mM each NTP, and 2 U enzyme in 1X buffer supplied by the manufacturer, at 4 °C overnight. The full-length product was purified by 7 M urea, 8% 29:1 mono:bis-polyacrylamide gel electrophoresis (PAGE) with the "crush-and-soak" method of gel

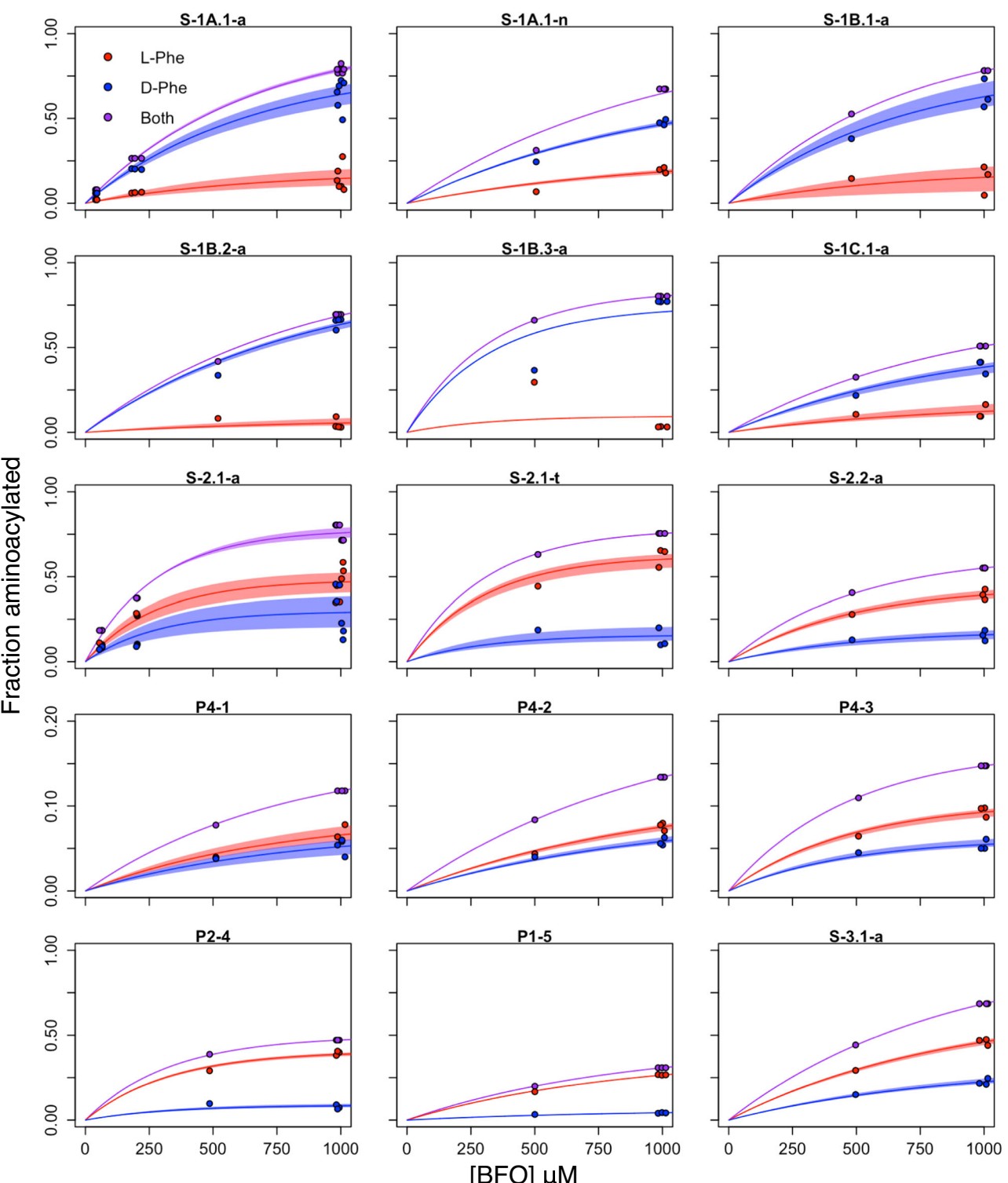

**Fig. 2 | Determination of reaction rates for L- and D- products for 15 ribozymes.** For each ribozyme, the fraction of RNA aminoacylated by L- (red) and D- (blue) BFO was measured at multiple concentrations of BFO and fit to the pseudo-first-order rate law as described, to estimate the enantioselectivity index $i_e = \log_2(k_D/k_L)$. Each point represents one experimental replicate. Replicates are jittered horizontally for visibility. Lines show the best fits to the rate law. Shaded areas indicate 95% confidence intervals from bootstrapping analysis. Source data are provided as a Source Data file. BFO = biotinyl-Phe-5(4H)-oxazolone.

extraction[48,49], followed by ethanol precipitation[50]. Purified RNA was dried in a vacuum centrifuge for 1.5 h and redissolved in TE buffer (10 mM Tris pH 8, 1 mM EDTA) and stored at −20 °C for up to 1 year. The purity of the full-length product was verified by PAGE with the same conditions.

L-RNA of ribozyme S-1A.1-a was purchased from Bio-Synthesis, Inc. including HPLC purification. The lyophilized pellet was resuspended in 10 mM Tris buffer (pH 8) and stored at −20 °C for up to one month.

Synthesis of substrate biotinyl-Phe-oxazolone (BFO) was performed according to a previously described procedure[17,34]. Reagents

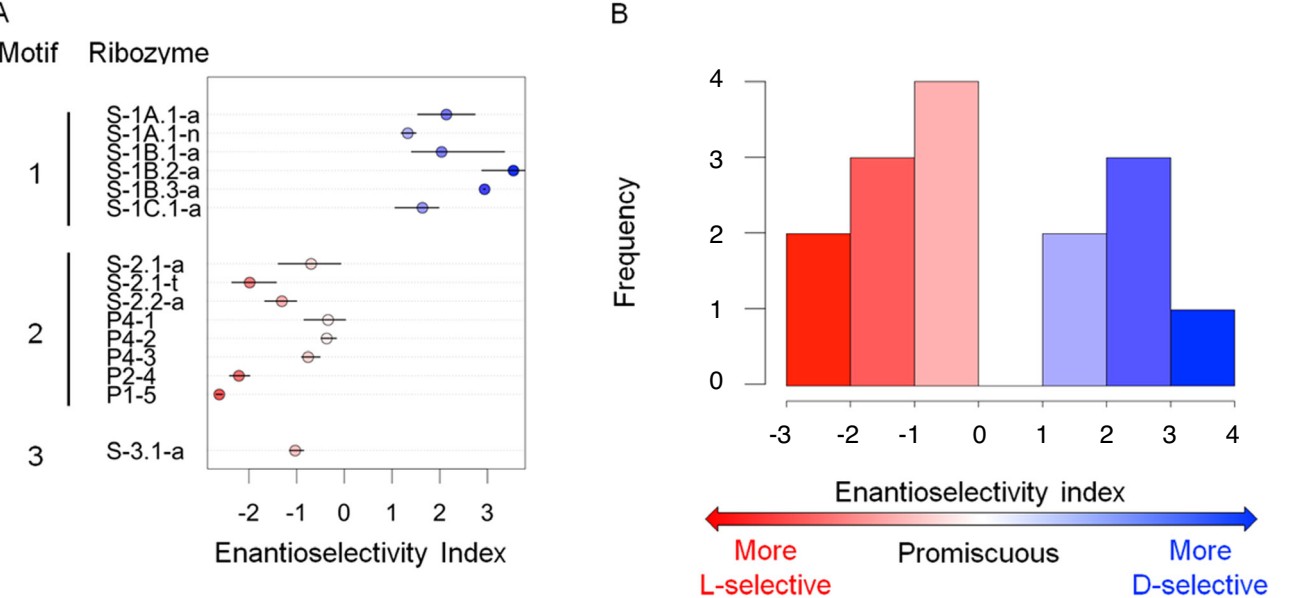

**Fig. 3 | Enantioselectivity indices for 15 ribozymes. A** Points show the median enantioselectivity index for each ribozyme listed. Error bars indicate 95% confidence intervals determined by bootstrapping the replicates from independent experiments (see Fig. 2) 1000 times. **B** The distribution of indices shows a range of positive and negative values, indicating both D- and L- selective ribozymes.

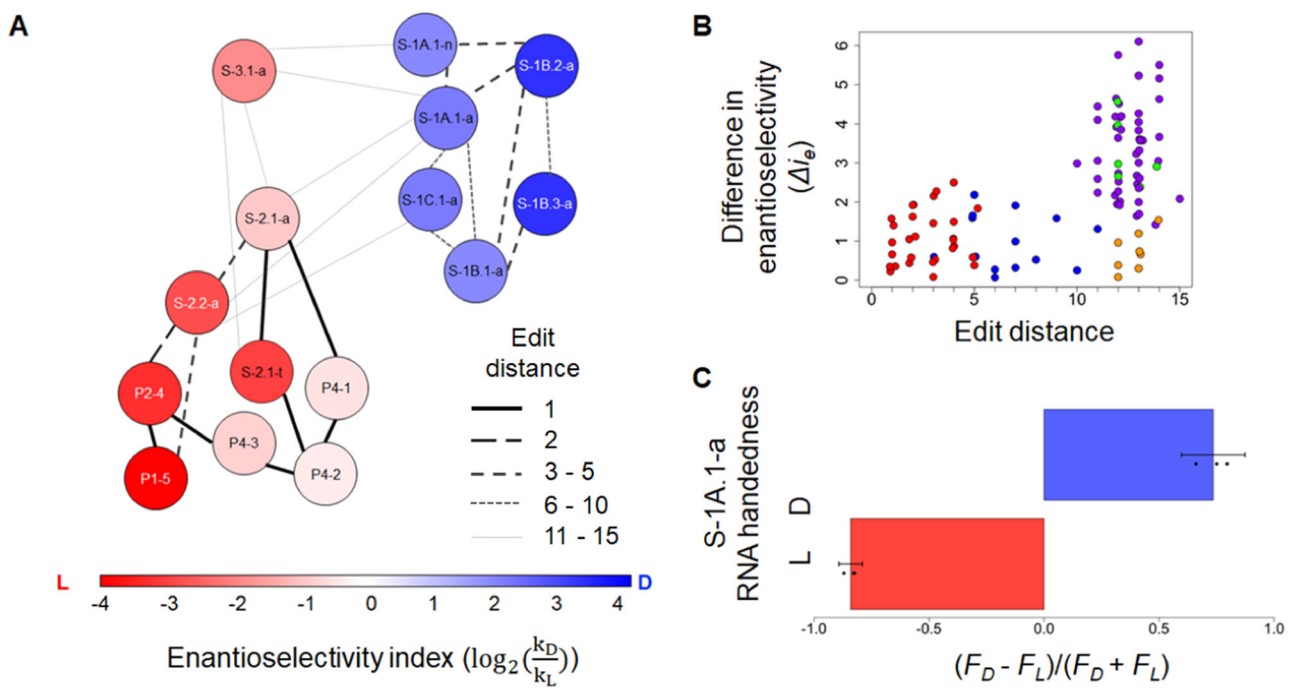

**Fig. 4 | Sequence relationships and stereoselectivity of ribozymes. A** The enantioselectivity index is shown as a heat map, with each ribozyme represented by a labeled circle. The Levenshtein edit distance (number of substitutions, insertions, or deletions) between sequences is indicated by the line strength (see legend). Direction of preference is conserved among related ribozymes. **B** Pairwise differences in enantioselectivity vs. edit distance for each pair of ribozymes. Blue: between ribozymes within Motif 1; red: between ribozymes within Motif 2; purple: between ribozymes of Motifs 1 and 2; green: between ribozymes of Motifs 1 and 3; orange: between ribozymes of Motifs 2 and 3. **C** Stereoselectivity of D- vs. L-ribozyme S-1A.1-a. Bars indicate the mean of triplicates; error bars indicate the standard error of the mean. The enantiomeric excesses [ee = |$(F_D − F_L)/(F_D + F_L)$|], were not significantly different (two-sided $t$-test, $p = 0.11$). Source data are provided as a Source Data file. $F$ = fraction; $i_e$ = enantioselectivity index.

and solvents were obtained from Millipore Sigma and used without further purification. The identity and purity of the product were verified by [1]H NMR analyzed by MNova 14. Vacuum-dried solid BFO was dissolved in anhydrous acetonitrile with sonication, to a concentration of 25 mM, and stored at −20 °C for up to 3 months.

**Aminoacylation reaction**

To 6 mL of an aqueous buffer solution of 100 mM HEPES (pH 7) containing 100 mM NaCl, 100 mM KCl, 5 mM $MgCl_2$, and 5 mM $CaCl_2$, BFO stock solution was added to the desired concentration and mixed thoroughly. Then, RNA was added to a resultant concentration of

10 ng/µL (~400 nM). Because the reaction with the lowest [BFO] contained 50 µM BFO, all reactions contained >100-fold molar excess of substrate over RNA. The reaction was incubated on a rotator at room temperature overnight.

## Hydrolysis of reacted RNA to isolate biotinyl-phenylalanine

The reaction solution after incubation was concentrated to ~50 µL (Amicon Ultra-4 Centrifugal Filter Units, Millipore Sigma), after which the RNA was exchanged into 60 mM $Na_2B_4O_7$ buffer (adjusted to pH 9.5 with NaOH) by size exclusion chromatography (Bio-Gel P-30, Bio Rad), removing excess substrate. The sample was then incubated overnight at 45 °C, during which time the mildly alkaline buffer selectively hydrolyzed the aminoacyl ester bond to liberate biotinyl-Phe from the RNA[51]. A BFO control was also subjected to the same buffer condition at room temperature. Aliquots (1 µL) of the RNA were taken before and after hydrolysis and analyzed by electrophoretic mobility shift assay (EMSA). EMSAs were performed by mixing samples with 2 µM streptavidin (aqueous stock solution, New England Biolabs, N7021S), incubating at room temperature for 5 min, adding 6X native loading dye (New England Biolabs), and loading without heating into native 8% 29:1 mono:bis-PAGE cast and run with the Bio-Rad minigel system (300 V). Gels were stained with SYBR Gold (Thermo Fisher Scientific, S11494) and imaged with an Amersham Typhoon scanner (General Electric). Images were collected and analyzed by Amersham Typhoon 1.1.0.7 and ImageQuant TL v 8.1. Aminoacylated RNA migrates more slowly than unmodified RNA due to streptavidin binding, resulting in a lower-mobility band. Successful hydrolysis of biotinyl-Phe was verified by the disappearance of the high-MW band. The RNA was removed from the hydrolysis reaction by size-based filtration (Amicon Ultra Centrifugal Filter Units, 10 kDa, Millipore Sigma) to isolate biotinyl-Phe.

## Quantification of L- and D-phenylalanine by chiral chromatography

The concentrations of L- and D-phenylalanine were determined by two methods. In the first method, which was used for analysis unless otherwise specified, biotinyl-Phe samples were desiccated by vacuum centrifuge overnight, dissolved in 100 µL 6 M HCl, and incubated at 100 °C for 3 h to remove biotin[35]. Samples were then neutralized with equimolar NaOH prior to column separation. Samples were diluted 1:1 with acetonitrile and analyzed by an Agilent InfinityLab 1290 Infinity II Series liquid chromatography system coupled to an Agilent 6470 triple quadrupole mass spectrometer. L- and D-phenylalanine were separated on a 150 mm Agilent InfinityLab Poroshell 120 Chiral column at a flow rate of 0.5 mL/min with column temperature of 25 °C and a 2 µL injection volume. The isocratic gradient was executed using a mobile phase comprised of 70:30 20 mM ammonium formate (adjusted to pH 3 with HCl): LC-MS grade methanol. Electrospray ionization was used, and the mass spectrometer source parameters were optimized to include a 330 °C $N_2$ gas temperature at a flow rate of 13 L/min, sheath gas temperature of 390 °C at a 12 L/min flow rate, a nebulizer ($N_2$) at 35 PSI, a capillary voltage of 1.5 kV and the nozzle voltage set at 0 V. The mass spectrometer was operated in positive ion mode and the mass analysis was performed under multiple reaction monitoring mode. The total run time for each sample was 10 min. Needle wash with isopropyl alcohol was done between injections. The mass transition (precursor ion and fragment ion pairs) for both enantiomers of phenylalanine was 166–120 m/z with positive polarity, with retention times of 5 min for L-Phe and 5.8 min for D-Phe. Six-point calibration standards ranging from 1 ng/mL to 100 ng/mL were used for absolute quantitation.

An analytical technique based on previously validated methods[35,52,53] was used to verify the accuracy of the first method described above. In this second method, biotinyl-Phe samples were desiccated under vacuum at room temperature and subjected to acid vapor hydrolysis at 150 °C for 3 h, using 6 M double-distilled hydrochloric acid to remove biotin[35,52]. Samples were then desalted by cation exchange chromatography[51] and prepared for analysis by pre-column derivatization with o-phthaldialdehyde/N-acetyl-L-cysteine (OPA/NAC)[50]. OPA/NAC is a chiral, fluorescent tag that enhances analytical sensitivity and specificity for primary amino groups and facilitates chromatographic separation of chiral amino acid enantiomers. Sample desalting prior to derivatization was necessary because the presence of interfering ions, such as salts, can hinder the OPA/NAC derivatization of amino acids[54]. Racemic tryptophan (Trp) was used as an internal standard during execution of ion exchange chromatography to account for potential loss of phenylalanine during the desalting process. Tryptophan was chosen as the internal standard because it shared the aromatic, α-amino chemical properties also featured by phenylalanine, thus making tryptophan an appropriate proxy by which to evaluate the possible likelihood of phenylalanine loss during ion exchange chromatography. Sample analysis was performed using a Thermo Fisher Scientific Accela 1250 ultra-high performance liquid chromatograph and a Thermo Fisher Scientific Accela autosampler in combination with a Dionex UltiMate 3000 RS Fluorescence Detector and a Thermo Fisher Scientific LTQ Orbitrap XL high-resolution mass spectrometer. Chromatography of target analytes was achieved using a 150 mm Waters ACQUITY CSH Phenyl Hexyl column, a 100 mm Waters ACQUITY CSH C18 column, and a 150 mm Waters ACQUITY CSH Phenyl Hexyl column in series. Analytes were eluted using an isocratic flow (130 µL/min) of 60:40 aqueous:organic, where the aqueous eluent was 45 mM ammonium formate (pH 9.0) with 7% LC-MS grade methanol, and the organic eluent was LC-MS grade methanol. The aqueous eluent was prepared by mixing 1.51 mL of formic acid in 780 mL of ultrapure water before titrating the solution to pH 9.0 using 1 M aqueous ammonium hydroxide, dropwise, before finally adding 64 mL of LC-MS methanol to the eluent. The 1 M ammonium hydroxide solution was prepared by diluting a 7.6 M stock solution of aqueous ammonium hydroxide (assay = 29.7 %, ammonia in water) with ultrapure water until a concentration of 1 M was achieved. The chromatography columns were maintained at 40 °C, the autosampler was kept at 25 °C, the injection volume was 10 µL, and the run time was 25 minutes. The fluorescence detector was operated with 340 nm excitation and 450 nm emission wavelengths. Given that OPA/NAC amino acid derivatives have finite stabilities[55], each standard, blank, control, and sample was immediately stored in a −80 °C freezer after injection to mitigate the degradation of amino acid derivatives and preserve these derivatives for the purpose of performing replicate analyses.

The Orbitrap mass spectrometer used in this second method was employed in concert with an electrospray ionization source operated in positive ion mode. Orbitrap data was collected in Full MS scan mode over a range of 300–800 m/z. The mass spectrometer front-end gas parameters included: sweep gas ($N_2$) flow rate of 1 a.u., sheath gas ($N_2$) flow rate of 40 a.u., and auxiliary gas ($N_2$) flow rate of 5 a.u. Ion source parameters used included a source voltage of 4.0 kV, a capillary temperature and voltage of 275 °C and 31 V, respectively, tube lens voltage of 85 V, and a skimmer offset voltage of 0.0 V. The Orbitrap mass analyzer was operated at a resolution setting of 60,000 (at full-width-half-maximum for m/z 400) with an automatic gain control target of $5 \times 10^5$ ions and a maximum ion time of 100 ms, while the number of microscans was set at 1. Lastly, the mass spectrometer was calibrated over the 150–2000 m/z range using the Thermo Scientific Pierce LTQ ESI positive ion calibration solution, which was composed of Ultramark 1621, caffeine, and MRFA (Met-Arg-Phe-Ala) in an acetic acid/methanol/acetonitrile solution. Additionally, accurate mass analysis was facilitated using a polysiloxane compound (m/z 371.10123, $[(C_2H_6SiO)_5 + H]^+$) found in ambient air as an internal lock mass, which resulted in a typical mass accuracy of <1 ppm. Thermo Fisher Scientific Xcalibur version 2.1 and Agilent Masshunter 8.0 were used for data collection.

## Determination of enantioselectivity index

We measured the chiral output (L-Phe vs. D-Phe) for experimental replicates of reaction with each ribozyme, at multiple substrate

concentrations and with overnight incubations, as described above. The fraction of total RNA aminoacylated by each stereoisomer in each reaction was calculated as the product of the fraction of L or D-Phe [$c_L/(c_L + c_D)$ or $c_D/(c_L + c_D)$, where $c_X$ is the concentration of isomer $X$ measured by LC-MS], and the fraction of total RNA aminoacylated, as measured by EMSA. To calculate the ratio of the rate constants ($k_D/k_L$), a kinetic model was developed (Supplementary Note 1). The total aminoacylation reaction was assumed to follow pseudo-first-order kinetics with [BFO] » [RNA], and [BFO] was assumed to be constant[17,34]. Since the substrate epimerizes[33,56] and was confirmed to contain a 50:50 mixture of D and L isomers (see "Results"), and [BFO] » [RNA], [D] was assumed to be equal to [L]. In this model, the integrated rate equation is $F_X([BFO]) = \frac{Ak_X}{k_L + k_D}(1 - e^{-[BFO](k_L + k_D)t})$, where $F_X$ is the fraction of total RNA aminoacylated with the $X$-Phe isomer ($X$ = D or L), $A$ is the maximum fraction of total RNA aminoacylated, and $t$ is the incubation time (in these experiments, $t = 16$–20 h). To determine the ratio $k_D/k_L$, first, the total amount of aminoacylated RNA ($F_D + F_L$) was fit to the form $[C(1 - e^{-[BFO](k_L + k_D)t})]$ to estimate $k_D + k_L$. Then, this value of $k_D + k_L$ was fixed in the integrated rate equation, and data for $F_X$ over [BFO] was fit to estimate $Ak_X$ for each isomer, allowing calculation of $k_D/k_L$ for each ribozyme. Note that $k_D/k_L$ is insensitive to the choice of reaction time $t$ (Supplementary Note 1). The log of this ratio [$\log_2(k_D/k_L)$], here termed the enantioselectivity index ($i_e$), was used to characterize the selectivity of each ribozyme. Errors were calculated as 95% confidence intervals, determined by a bootstrapping procedure sampling replicate measurements with replacement and calculating $k_D/k_L$ ($n = 1000$). Data were analyzed using Microsoft Excel v 16 and R version 4.1.

### Reporting summary
Further information on research design is available in the Nature Portfolio Reporting Summary linked to this article.

## Data availability
The authors declare that the data supporting the findings of this study are available within the paper and its supplementary information files. Source data are provided with this paper.

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

## Acknowledgements

We thank Yang Li and Xiangning Huang for assistance with HPLC-MS and Donna Blackmond and John Sutherland for discussions. NMR was performed in the MRL Shared Experimental Facilities, which are supported by the MRSEC Program of the NSF under Award No. DMR 1720256; a member of the NSF-funded Materials Research Facilities Network (www.mrfn.org). Funding from NASA (NNX16AJ32G and 80NSSC21K0595), the Simons Foundation Collaboration on the Origin of Life (290356FY18 to I.A.C. and 302497 to J.P.D.), NSF (EF 1935372, MCB 2318736) and the UCSB Mellichamp Graduate Fellowship in Systems Biology and Bioengineering are acknowledged. A.V.-S.'s research was supported by an appointment to the NASA Postdoctoral Program, administered by Oak Ridge Associated Universities under contract with NASA.

## Author contributions

J.K. and I.A.C. conceived the project. J.K., A.V.S., R.W., K.B., E.J., and K.M.S. conducted experiments and analyzed data. Z.L. synthesized and characterized BFO substrate. J.K., A.V.S., W.L., E.T.P., and J.P.D. performed chiral analysis. J.K. and I.A.C. developed the model for data fitting. J.K. wrote the first draft of the manuscript. A.V.S., J.K., E.J., E.T.P., J.P.D., and I.A.C. edited the manuscript. All authors discussed the results throughout the project and approved the final manuscript. I.A.C. supervised the project.

## Competing interests

The authors declare no competing interests.
