## [Peer Review File · Nature Communications]

Prebiotic chiral transfer from self-aminoacylating ribozymes
may favor either handednessREVIEWER COMMENTS

Reviewer #1 (Remarks to the Author):

I read the article by Kenchel et al. with great interest. The authors showed that RNA may transfer chiral information to protein through substrate selectivity; and this selectivity does not result from the chirality of the ribose in ribozyme, but from the sequence/structure of the ribozyme. The finding of this paper is of fundamental significance, which implies that life may have had different choices in the chirality of its components. A further question naturally arises, why the current combination, i.e., D-ribose and L-amino acids, was chosen. Some specific comments are as follows.

1) This paper proposed that the chiral biological world may be generated “as random as a coin flip”. This is possible, but other possibilities should not be ignored. L-amino acids may be more abundant on primitive Earth as has been found in meteorites. Prior studies also showed that D-ribose RNA is inclined to bind to L-amino acids to a certain extent [1,2] and may be protected by the later [3]. These differences may bring the ribozymes that prefer L-amino acids an evolutionary advantage, and the initially slight differences can be amplified by the replication of ribozymes [3]. Accordingly, I suggest the authors to provide more discussions regarding this point in their text.

2) As implied in Figure 2, the catalytic activity of the ribozymes used in this study should also be different. A comparison of the activities of these ribozymes under conditions closer to the primitive Earth (e.g., lower concentration of substrate) will be of interest. If the activity of ribozymes is independent of their substrate preference, the choice of L-amino acids should be more likely to be random.

3) Because in prebiotic world, some amino acids were likely more prevalent than others [4], it would be better to use these amino acids (such as Ala, Glu, Ser) to investigate the chirality origin rather than using Phe, which is considered relatively rare in primitive world.

References

1. Root-Bernstein, R. (2010) Experimental test of l- and d-amino acids binding to l- and d-codons suggests that homochirality and codon directionality emerged with the genetic code. *Symmetry* 2: 1180-1200.
2. Illangasekare, M., Turk, R., Peterson, G.C., Lladser, M., and Yarus, M. (2010) Chiral

histidine selection by d-ribose RNA. RNA 16: 2370-2383.

3. Chu, X.Y., Zhang, H.Y. (2021) Protein homochirality may be derived from primitive peptide synthesis by RNA. Astrobiology 21: 628-635.

4. Zaia, D.A., Zaia, C.T.B., De Santana, H. (2008) Which amino acids should be used in prebiotic chemistry studies? Orig Life Evol Biosph 38: 469-488.

Reviewer #2 (Remarks to the Author):

In their manuscript “Prebiotic chiral transfer from self-aminoacylating ribozymes may favor either handedness”, Chen and co-workers probed the ability of self-aminoacylating D-ribozyme models to discriminate between the enantiomers of biotinyl-Phe-5(4H)-oxazolone (BFO), an activated phenylalanine derivative. The work is part of the fascinating area related to the elusive origin of the almost exclusive homochiral nature of extant life which mostly consists in finding the nature of the symmetry breaking event and chemical/physical fields that leads to its amplification. Here, the field is approached from a different angle: the authors probe a potential stereochemical relationship between aminoacylation ribozymes – as model of D-RNAs present in the RNA World – and the acylated amino acids issue from a racemic mixture of BFO. It is worth to note that only a handful of publications have addressed a potential stereochemical relationship between ribozymes and proteins. The working hypotheses and related interpretations are solid. A particular challenge in the context of the discrimination between enantiomers of amino acids with a chiral scaffold comes from potential contaminations or analytical inconsistencies that have led in the past to erroneous interpretations (Science, 1974, 143, and more recent work summarized by Meierhenrich in the Chapter 2 of U. Meierhenrich, Amino acids and the asymmetry of life: caught in the act of formation, Springer, Berlin, 2008.). Here, analytical methods appear to be solid and both methods used to quantify the enantiomeric preference have been cross-validated (Figure S4). The use of chiral LC/MS also further supports that the stereochemical discrimination recorded for these ribozymes is free of any analytical bias. The main result of this study is that various motifs of ribozymes exhibit varying levels of selectivity and, most importantly, that the direction of this selectivity can be opposite. I have only relatively minor points. Chiral preference in the literature is mostly reported under the form of enantiomeric excesses – while in the present case enantioselectivity indexes are provided

(that considers the amount of acylated RNA). I presume that, again based on the acylated fraction of BFO, data can be reported as enantiomeric excesses, which may be more convenient to compare different discriminating sources in the literature. In the discussion, the reported data are interpreted as “the ultimate emergence of an L- vs. D-protein world may have been as random as a coin flip.”; i.e. the authors include their work in the context of “by chance mechanisms” that could have led to biological homochirality. However, authors must admit that the present study only looks for a certain type of activated amino acid, complexes to RNA but not polymerized by, and that a by chance mechanism would suggest that Life has emerged from a single type of ribozymes. I suggest the authors to be more open to alternative theories such as the that proteins (“metabolism first” theory) or lipids originated first or that RNA, DNA and proteins emerged simultaneously by continuous and reciprocal interactions, i.e. mutualism. The possibility of having two “equal runners” in the primordial World, i.e. both enantiomer of homochiral polymers, has also attracted most interest in the recent years. Minor typo: references 31 and 33 are the same.

Response Letter

We thank the reviewers for their careful reading and valuable comments.

Reviewer #1 (Remarks to the Author):

I read the article by Kenchel et al. with great interest. The authors showed that RNA may transfer chiral information to protein through substrate selectivity; and this selectivity does not result from the chirality of the ribose in ribozyme, but from the sequence/structure of the ribozyme. The finding of this paper is of fundamental significance, which implies that life may have had different choices in the chirality of its components. A further question naturally arises, why the current combination, i.e., D-ribose and L-amino acids, was chosen. Some specific comments are as follows.

1) This paper proposed that the chiral biological world may be generated “as random as a coin flip”. This is possible, but other possibilities should not be ignored. L-amino acids may be more abundant on primitive Earth as has been found in meteorites. Prior studies also showed that D-ribose RNA is inclined to bind to L-amino acids to a certain extent [1,2] and may be protected by the later [3]. These differences may bring the ribozymes that prefer L-amino acids an evolutionary advantage, and the initially slight differences can be amplified by the replication of ribozymes [3]. Accordingly, I suggest the authors to provide more discussions regarding this point in their text.

We agree and indeed we have described the work in the references given by the reviewer (1-3) in the Introduction. We added new text to the Discussion to emphasize this point and clarify the distinction between our study, concerning chiral transfer, and other mechanisms that affect symmetry breaking or amplification processes.

2) As implied in Figure 2, the catalytic activity of the ribozymes used in this study should also be different. A comparison of the activities of these ribozymes under conditions closer to the primitive Earth (e.g., lower concentration of substrate) will be of interest. If the activity of ribozymes is independent of their substrate preference, the choice of L-amino acids should be more likely to be random.

The concern is whether D/L substrate preference is sensitive to substrate concentration, particularly low concentrations that might characterize the primitive Earth. The i_e reported in Table 1 were calculated from fitting data at multiple concentrations, thus integrating information from the maximum available data. However, i_e can also be calculated separately at each concentration, as described in Text S1. Therefore, to check this concern, we separately calculated i_e at each concentration and plotted the i_e of products against substrate concentration. This analysis was done for two ribozymes, S-1A.1-a and S-2.1-a. representing two different motifs with opposite preference, for four substrate concentrations (50 μ M to 1 mM). No systematic dependence was observed (new Figure S7; also new Source Data file).

The lack of dependence of i_e on substrate concentration is also consistent with a pseudo-first-order kinetic model of the ribozyme reaction, as described in Text S1.

3) Because in prebiotic world, some amino acids were likely more prevalent than others [4], it

would be better to use these amino acids (such as Ala, Glu, Ser) to investigate the chirality origin rather than using Phe, which is considered relatively rare in primitive world.

References

1. Root-Bernstein, R. (2010) *Experimental test of L- and D-amino acids binding to L- and D-codons suggests that homochirality and codon directionality emerged with the genetic code.* *Symmetry* 2: 1180-1200.
2. Illangasekare, M., Turk, R., Peterson, G.C., Lladser, M., and Yarus, M. (2010) *Chiral histidine selection by D-ribose RNA.* *RNA* 16: 2370-2383.
3. Chu, X.Y., Zhang, H.Y. (2021) *Protein homochirality may be derived from primitive peptide synthesis by RNA.* *Astrobiology* 21: 628-635.
4. Zaia, D.A., Zaia, C.T.B., De Santana, H. (2008) *Which amino acids should be used in prebiotic chemistry studies?* *Orig Life Evol Biosph* 38: 469-488.

While Phe has been found in meteorites, it is true that Phe is not considered to be very abundant in the prebiotic world. The decision to use an aromatic side chain in this analysis was essentially practical, because aromatic substrates react strongly with multiple different ribozymes (which were originally selected due to reaction with aromatic substrates). We have observed that some of the ribozymes also react promiscuously with substrates having small hydrophobic side chains (Val, Ile, Leu) (Janzen et al., *Nat Commun* 13:3631). However, of the ribozymes we have studied, only Motif 2 ribozymes react with the alternative substrates at a high enough level that would generate sufficient material for chiral LCMS analysis.

In principle, it would be possible to synthesize an alternative substrate and test the Motif 2 ribozymes for activity and chiral preference. However, the results would not be interpretable in terms of the larger question. For example, the probable result in which all the Motif 2 ribozymes share the same preference cannot be interpreted as an intrinsic preference of ribozymes in general, because only one motif is being sampled. This problem could be solved if we had multiple different ribozyme motifs that react with the alternative substrate. However, obtaining new motifs would require a new *in vitro* selection with the alternative substrate to discover the ribozymes as well as validation and characterization of individual sequences. The work required would be equivalent to the work that previously constituted an entire paper itself (in particular, our original selection and ribozyme discovery in Pressman et al., *J Am Chem Soc* 141:6213-6223). On top of this, there is also a high risk of failure for *in vitro* selection itself as a technique. Therefore, unfortunately, I do not judge this to be a reasonable undertaking for my group.

Nevertheless, the point that Phe was used here as part of a model reaction, not as a prebiotically likely early amino acid, is important and well-taken. We added text to the Discussion (paragraph 1) to point out this caveat, including reference 4 from the reviewer.

Reviewer #2 (Remarks to the Author):

In their manuscript "Prebiotic chiral transfer from self-aminoacylating ribozymes may favor either handedness", Chen and co-workers probed the ability of self-aminoacylating D-ribozyme models to discriminate between the enantiomers of biotinyI-Phe-5(4H)-oxazolone (BFO), an activated phenylalanine derivative. The work is part of the fascinating area related to the elusive origin of the almost exclusive homochiral nature of extant life which mostly consists in finding the nature of the symmetry breaking event and chemical/physical fields that leads to its amplification. Here, the field is approached from a different angle: the authors probe a potential

stereochemical relationship between aminoacylation ribozymes – as model of D-RNAs present in the RNA World – and the acylated amino acids issue from a racemic mixture of BFO. It is worth to note that only a handful of publications have addressed a potential stereochemical relationship between ribozymes and proteins. The working hypotheses and related interpretations are solid. A particular challenge in the context of the discrimination between enantiomers of amino acids with a chiral scaffold comes from potential contaminations or analytical inconsistencies that have led in the past to erroneous interpretations (Science, 1974, 143, and more recent work summarized by Meierhenrich in the Chapter 2 of U. Meierhenrich, Amino acids and the asymmetry of life: caught in the act of formation, Springer, Berlin, 2008.).

This useful reference was added.

Here, analytical methods appear to be solid and both methods used to quantify the enantiomeric preference have been cross-validated (Figure S4). The use of chiral LC/MS also further supports that the stereochemical discrimination recorded for these ribozymes is free of any analytical bias. The main result of this study is that various motifs of ribozymes exhibit varying levels of selectivity and, most importantly, that the direction of this selectivity can be opposite. I have only relatively minor points. Chiral preference in the literature is mostly reported under the form of enantiomeric excesses – while in the present case enantioselectivity indexes are provided (that considers the amount of acylated RNA). I presume that, again based on the acylated fraction of BFO, data can be reported as enantiomeric excesses, which may be more convenient to compare different discriminating sources in the literature.

We have added a column to Table 1 with enantiomeric excess values. These ee data are also given in the added Source Data file. We preferred the enantioselectivity index i_e because of its straightforward relationship to the pseudo-first-order kinetic model for our reaction, but it is true that ee would be easier for comparison to other literature.

In the discussion, the reported data are interpreted as “the ultimate emergence of an L- vs. D-protein world may have been as random as a coin flip.”; i.e. the authors include their work in the context of “by chance mechanisms” that could have led to biological homochirality. However, authors must admit that the present study only looks for a certain type of activated amino acid, complexes to RNA but not polymerized by, and that a by chance mechanism would suggest that Life has emerged from a single type of ribozymes. I suggest the authors to be more open to alternative theories such as the that proteins (“metabolism first” theory) or lipids originated first or that RNA, DNA and proteins emerged simultaneously by continuous and reciprocal interactions, i.e. mutualism. The possibility of having two “equal runners” in the primordial World, i.e. both enantiomer of homochiral polymers, has also attracted most interest in the recent years.

We added text to the Discussion (last paragraph) to include this general caveat that other chemical principles might operate in other origin scenarios.

Minor typo: references 31 and 33 are the same.

The typo was fixed.

REVIEWERS' COMMENTS

Reviewer #1 (Remarks to the Author):

I think the authors have addressed my concerns about their article, so I support the publication of this paper in Nature Communications.

Reviewer #2 (Remarks to the Author):

The revised version of the paper "Prebiotic chiral transfer from self-aminoacylating ribozymes may favor either handedness" incorporates the requested changes, notably concerning the enantiomeric excesses that can be expected from the acylation process and a more detailed discussion about the integration of the present work regarding on the elusive question of the origin of biological homochirality. The authors just need to precise how the enantiomeric excesses are extracted from the exp. data and why there are quite low relatively to the enantiomeric excesses measured by HPLC-MS. The comparison of HPLC traces in Figure 1 and e.e. values in Table 1 can be quite confusing for the reader. It is presumably due to excess of substrates used in the acylation, but this should be mentioned to avoid confusion.

Response to Reviewers (second revision)

Reviewer #1 (Remarks to the Author):

I think the authors have addressed my concerns about their article, so I support the publication of this paper in Nature Communications.

Reviewer #2 (Remarks to the Author):

The revised version of the paper "Prebiotic chiral transfer from self-aminoacylating ribozymes may favor either handedness" incorporates the requested changes, notably concerning the enantiomeric excesses that can be expected from the acylation process and a more detailed discussion about the integration of the present work regarding on the elusive question of the origin of biological homochirality. The authors just need to precise how the enantiomeric excesses are extracted from the exp. data and why there are quite low relatively to the enantiomeric excesses measured by HPLC-MS. The comparison of HPLC traces in Figure 1 and e.e. values in Table 1 can be quite confusing for the reader. It is presumably due to excess of substrates used in the acylation, but this should be mentioned to avoid confusion.

We realized that the % ee was actually given in Table 1 as a fraction; thus the values may have appeared low. We are sorry for the confusion. We changed the units to percentages. We also added text to the caption of Table 1 to illustrate the ee calculation, using the values from the HPLC trace shown in Figure 1D for ribozyme S-1A.1-a. To make this figure consistent, we now show the gels for ribozyme S-1A.1-a as the example for Figure 1A, so the reader can follow the same reaction through this figure.